# A Continuous-Time Mirror Descent Approach to Sparse Phase Retrieval

**Fan Wu**
Department of Statistics
University of Oxford
`fan.wu@stats.ox.ac.uk`

**Patrick Rebeschini**
Department of Statistics
University of Oxford
`patrick.rebeschini@stats.ox.ac.uk`

## Abstract

We analyze continuous-time mirror descent applied to sparse phase retrieval, which is the problem of recovering sparse signals from a set of magnitude-only measurements. We apply mirror descent to the unconstrained empirical risk minimization problem (batch setting), using the square loss and square measurements. We provide a convergence analysis of the algorithm in this non-convex setting and prove that, with the hypentropy mirror map, mirror descent recovers any $k$-sparse vector $\mathbf{x}^\star \in \mathbb{R}^n$ with minimum (in modulus) non-zero entry on the order of $\|\mathbf{x}^\star\|_2/\sqrt{k}$ from $k^2$ Gaussian measurements, modulo logarithmic terms. This yields a simple algorithm which, unlike most existing approaches to sparse phase retrieval, adapts to the sparsity level, without including thresholding steps or adding regularization terms. Our results also provide a principled theoretical understanding for Hadamard Wirtinger flow [54], as Euclidean gradient descent applied to the empirical risk problem with Hadamard parametrization can be recovered as a first-order approximation to mirror descent in discrete time.[1]

## 1 Introduction

Mirror descent [37] is becoming increasingly popular in a variety of settings in optimization and machine learning. One reason for its success is the fact that mirror descent can be adapted to fit the geometry of the optimization problem at hand by choosing a suitable strictly convex potential function, the so-called mirror map. Mirror descent has been extensively studied for convex problems, and it is amenable to a general convergence analysis in terms of the Bregman divergence associated to the mirror map, e.g. [5, 6, 7, 8, 12, 36, 44]. There is a growing literature considering mirror descent in non-convex settings, e.g. [18, 16, 20, 21, 29, 30, 34, 57, 59, 60], and we contribute to this literature by analyzing continuous-time mirror descent in the non-convex problem of sparse phase retrieval.

Recently, there has been a surge of interest in investigating continuous-time solvers in a variety of settings in machine learning, for instance, in connection to implicit regularization, e.g. [1, 2, 46], learning neural networks, e.g. [14, 35, 41, 45], and, more in general, to understand the foundations of algorithmic paradigms and to provide design principles for discrete-time algorithms used in practice, e.g. [3, 24, 40, 48, 52]. Convergence analyses for continuous-time algorithms are typically simpler and more transparent than those for their discrete-time counterparts, as they allow to focus on the main properties of the algorithm.

Phase retrieval is the problem of recovering a signal from the (squared) magnitude of a set of linear measurements. Such a task arises in many applications such as optics [49], diffraction imaging [9] and quantum mechanics [15], where detectors are able to measure intensities, but not phases. Due to the missing phase information, exploiting additional prior information often becomes necessary to

ensure that the problem is well-posed. Common forms of prior information include assumptions on sparsity, non-negativity or the magnitude of the signal [19, 27]. Other approaches include introducing redundancy by oversampling random Gaussian measurements or coded diffraction patterns [11, 13].

Numerous strategies have been developed to exploit sparsity. One approach is to confine the search to the low-dimensional subspace of sparse vectors, either via a preliminary support recovery step as in the alternating minimization algorithm SparseAltMinPhase [38] or by updating the current estimated support in a greedy fashion as in GESPAR [42]. Another approach relies on the introduction of thresholding steps to enforce sparsity. This approach is typically found in non-convex optimization based algorithms such as thresholded Wirtinger flow (TWF) [10], sparse truncated amplitude flow (SPARTA) [51], compressive reweighted amplitude flow (CRAF) [56], and sparse Wirtinger flow (SWF) [55]. Sparsity can also be promoted by augmenting the objective function with a regularization term. This is the approach taken in convex relaxation based methods such as compressive phase retrieval via lifting (CPRL) [39] and SparsePhaseMax [25], which include an $\ell_1$ penalty, but also in PR-GAMP [43], which is an algorithm based on generalized message passing that uses a sparsity inducing prior. Hadamard Wirtinger flow (HWF) [54] is an algorithm which performs gradient descent on the unregularized empirical risk using the Hadamard parametrization. This parametrization has been recently used to recover low-rank structures in sparse recovery [26, 47, 58] and matrix factorization [4, 23, 32] under the restricted isometry property.

With the exception of HWF, the aforementioned methods rely on restricting the search to sparse vectors, thresholding steps or adding regularization terms to enforce sparsity. On the other hand, HWF does not require thresholding steps or added regularization terms to promote sparsity. Further, it has been empirically observed that HWF can recover sparse signals from a number of measurements comparable to those required by PR-GAMP, in particular, requiring fewer measurements than existing gradient based algorithms. Despite these benefits, the work introducing HWF in [54] has two main limitations: on the one hand, a full theoretical understanding of the algorithm that can explain its (convergence) behavior and, in particular, the reason why it adapts to the signal sparsity, is lacking. On the other hand, the algorithm has been empirically shown to have a sublinear convergence phase to the underlying signal, which would seem to lead to improved sample complexity at the expense of an increase in computational cost.

## 1.1 Our contributions

In this work, we provide a theoretical analysis of unconstrained mirror descent in continuous time applied to the unregularized empirical risk with the square loss for the problem of sparse phase retrieval with square measurements. With the hypentropy mirror map [22], we prove that mirror descent recovers any $k$-sparse vector $\mathbf{x}^\star \in \mathbb{R}^n$ with minimum non-zero entry (in modulus) on the order of $\|\mathbf{x}^\star\|_2/\sqrt{k}$ from $\widetilde{\mathcal{O}}(k^2)$ Gaussian measurements, where $\widetilde{\mathcal{O}}$ hides logarithmic terms. To the best of our knowledge, this is the first result on continuous-time solvers for (sparse) phase retrieval.

This provides a simple first-order method that relies neither on thresholding steps, nor on regularized objective functions. Without requiring knowledge of the sparsity level $k$, mirror descent *adapts* to the sparsity level via the geometry defined by the hypentropy mirror map. This mirror map is parametrized by $\beta$, which is the only parameter in mirror descent and regulates the magnitude that off-support variables can attain while the algorithm runs. Our analysis shows that $\beta$ should be chosen smaller than a quantity depending on the signal size $\|\mathbf{x}^\star\|_2$ and the ambient dimension $n$. In particular, tuning of $\beta$ does *not* require knowledge (or estimation) of the sparsity level $k$. We remark that estimating the signal size $\|\mathbf{x}^\star\|_2$ is easily done by considering the average observation size [11, 50]. We initialize mirror descent following the same initialization scheme proposed in [54] for HWF. This initialization is independent of $\beta$ and only requires knowledge of a single coordinate on the support, which can be estimated from $\widetilde{\mathcal{O}}(k^2)$ Gaussian measurements [54]. This initialization is much simpler than the schemes typically necessary for other non-convex formulations such as the spectral initialization in CRAF or the orthogonality-promoting initialization in SPARTA.

It was observed in [48] that gradient descent with the Hadamard parametrization can be seen as a first-order approximation to mirror descent with the hypentropy mirror map. Since HWF consists of running vanilla gradient descent on the unregularized empirical risk under the Hadamard parametrization, it can be treated as a discrete-time first-order approximation to the mirror descent algorithm we analyze. Hence, our work provides a principled theoretical understanding for HWF and addresses the first of the two limitations mentioned above on the analysis given in [54]. Further, our

investigation also reveals the connection between the initialization size in HWF (which corresponds to the mirror map parameter $\beta$ up to a squareroot) and the convergence speed of the algorithm. In particular, we first have an initial warm-up period, after which convergence towards the true signal $\mathbf{x}^\star$ is linear, up to a precision determined by the parameter $\beta$ and the signal dimension $n$. By choosing a sufficiently small initialization, any desired accuracy can be reached before entering the final sublinear stage, which is the main reason for the slow convergence of HWF observed in [54].

The property that enables mirror descent to deal with the non-convexity of the objective function is a weaker version of *variational coherence* [59, 60]. While the variational coherence property as defined in [59, 60] precludes the existence of saddle points and is not satisfied in the sparse phase retrieval problem, we show that the defining inequality is satisfied along the trajectory of mirror descent, which is what allows us to establish the convergence analysis.

The literature on mirror descent is vast and growing, and a full overview is outside the scope of our work. Our contribution adds, in particular, to existing results on mirror descent in non-convex settings, which in general either require specific assumptions [16, 59, 60], guarantee convergence only to stationary points [17, 18, 20, 21, 57], or are tailored to specific problems in the online learning setting [29, 30, 34], for instance. Recently, the connection between sparsity and mirror descent equipped with the hypentropy mirror map was established in [48]. While the aforementioned connection has been shown for linear models and kernels, which leads to a convex problem, our analysis demonstrates that this connection also extends to the non-convex problem of sparse phase retrieval.

## 2 Preliminaries

We first introduce some notation. We use boldface letters for vectors and matrices, normal font for real numbers, and, typically, uppercase letters for random and lowercase letters for deterministic quantities. For the clarity of the analytical results, this paper focuses on sparse phase retrieval with real signal and measurement vectors. Nevertheless, the algorithm also works in the complex case.

### 2.1 Mirror descent

We first give a brief description of the unconstrained mirror descent algorithm; more details can be found in [8]. The key object defining the geometry of the algorithm is the *mirror map*.

**Definition 1** *Let $\mathcal{D} \subset \mathbb{R}^n$ be a convex open set. We say that $\Phi : \mathcal{D} \to \mathbb{R}$ is a mirror map if it is strictly convex, differentiable and its gradient takes all possible values, i.e. $\{\nabla\Phi(\mathbf{x}) : \mathbf{x} \in \mathcal{D}\} = \mathbb{R}^n$.*

We consider unconstrained mirror descent, i.e. $\mathcal{D} = \mathbb{R}^n$. Let $F : \mathbb{R}^n \to \mathbb{R}$ be a (possibly non-convex) function, for which we seek a global minimizer. Mirror descent is characterized by the mirror map $\Phi$ and an initial point $\mathbf{X}(0)$, and is, in continuous time, defined by the identity [37]

$$\frac{d}{dt}\mathbf{X}(t) = -\left(\nabla^2\Phi(\mathbf{X}(t))\right)^{-1}\nabla F(\mathbf{X}(t)). \tag{1}$$

An important quantity in the analysis of mirror descent is the *Bregman divergence* associated to a mirror map $\Phi$, which is given by

$$D_\Phi(\mathbf{x}, \mathbf{y}) = \Phi(\mathbf{x}) - \Phi(\mathbf{y}) - \nabla\Phi(\mathbf{y})^T(\mathbf{x} - \mathbf{y}).$$

The following equality can be derived from a quick calculation:

$$\frac{d}{dt}D_\Phi(\mathbf{x}', \mathbf{X}(t)) = \langle \nabla F(\mathbf{X}(t)), \mathbf{x}' - \mathbf{X}(t)\rangle, \tag{2}$$

where $\mathbf{x}' \in \mathbb{R}^n$ is any reference point. In particular, when the objective function $F$ is convex, equation (2) shows that mirror descent monotonically decreases the Bregman divergence to any minimizer of $F$. In non-convex settings, the inner product in (2) was used to define the notion of variational coherence, which is the assumption under which convergence of a stochastic version of mirror descent towards a minimizer of $F$ has been shown [59, 60].

### 2.2 Sparse phase retrieval

The goal in sparse phase retrieval is to reconstruct an unknown $k$-sparse vector $\mathbf{x}^\star \in \mathbb{R}^n$ from a set of quadratic measurements $Y_j = (\mathbf{A}_j^T\mathbf{x}^\star)^2$, $j = 1, ..., m$, where the measurement vectors $\mathbf{A}_j \sim \mathcal{N}(0, \mathbf{I}_n)$ are i.i.d. and observed.

A well-established approach to estimating the signal $\mathbf{x}^\star$ is based on non-convex optimization [10, 51, 55, 56]. In particular, given observations $(\mathbf{A}_1, Y_1), ..., (\mathbf{A}_m, Y_m)$, the goal becomes minimizing the (non-convex) empirical risk

$$F(\mathbf{x}) = \frac{1}{4m} \sum_{j=1}^{m} \left( (\mathbf{A}_j^T \mathbf{x})^2 - Y_j \right)^2. \tag{3}$$

It is worth mentioning that a different, amplitude-based risk function has also been considered [51, 56]. However, in that case the objective function becomes non-smooth, as the terms $(\mathbf{A}_j^T \mathbf{x})^2$ are replaced with $|\mathbf{A}_j^T \mathbf{x}|$, and the analysis via mirror descent appears more challenging.

As a non-convex function, the function $F$ in (3) could potentially have many local minima and saddle points, and even global minima different from $\mathbf{x}^\star$. It has been shown that if we have $m \geq 4k - 1$ Gaussian measurements, then, with high probability, $\mathbf{x}^\star$ is (up to a global sign) the sparsest minimizer of $F$, that is $\{\pm \mathbf{x}^\star\} = \mathrm{argmin}_{\mathbf{x}: \|\mathbf{x}\|_0 \leq k} F(\mathbf{x})$ [31]. In order to tackle these difficulties, previous methods such as SPARTA [51] and CRAF [56] employed a sophisticated spectral or orthogonality-promoting initialization scheme, which produces an initial estimate close enough to the signal $\mathbf{x}^\star$, followed by thresholded gradient descent updates, which confine the iterates to the low-dimensional subspace of $k$-sparse vectors.

## 3 The algorithm

We consider unconstrained mirror descent in continuous time given by (1) and applied to the objective function (3), equipped with the mirror map [22]

$$\Phi(\mathbf{x}) = \sum_{i=1}^{n} \left( x_i \operatorname{arcsinh} \left( \frac{x_i}{\beta} \right) - \sqrt{x_i^2 + \beta^2} \right), \tag{4}$$

for some parameter $\beta > 0$. A discussion on the choice of the parameter $\beta$ is given in Section 4. The Hessian is given by the diagonal matrix with entries $\nabla^2 \Phi(\mathbf{x})_{ii} = (x_i^2 + \beta^2)^{-1/2}$.

For the initialization, following the approach outlined in [54], we set

$$X_i(0) = \begin{cases} \frac{1}{\sqrt{3}} \cdot \sqrt{\frac{1}{m} \sum_{j=1}^{m} Y_j} & i = i_0 \\ 0 & i \neq i_0 \end{cases} \tag{5}$$

for a coordinate $i_0$ on the support of the signal $\mathbf{x}^\star$, i.e. $x_{i_0}^\star \neq 0$. Here, the term $\sqrt{\frac{1}{m} \sum_{j=1}^{m} Y_j}$ is an estimator for the magnitude $\|\mathbf{x}^\star\|_2$, see e.g. [11, 50]. By Lemma 1 of [54], it is possible to estimate a coordinate in the true support with high probability from $\widetilde{\mathcal{O}}(k(x_{max}^\star)^{-2})$ Gaussian measurements, where $x_{max}^\star = \max_i |x_i^\star| / \|\mathbf{x}^\star\|_2$. We develop theory for mirror descent with $\widetilde{\mathcal{O}}(k^2)$ samples, which is the worst case of the bound $\widetilde{\mathcal{O}}(k(x_{max}^\star)^{-2})$, as $\mathbf{x}^\star$ is a $k$-sparse vector, and hence $x_{max}^\star \geq 1/\sqrt{k}$.

In order to gain some intuitive understanding of why the initialization (5) is suitable, it will be helpful to consider the limiting case $m = \infty$. The following explanation to motivate the choice of initialization is taken from [54]. The gradient of the empirical risk is given by

$$\nabla F(\mathbf{x}) = \frac{1}{m} \sum_{j=1}^{m} \left( (\mathbf{A}_j^T \mathbf{x})^2 - (\mathbf{A}_j^T \mathbf{x}^\star)^2 \right) (\mathbf{A}_j^T \mathbf{x}) \mathbf{A}_j. \tag{6}$$

A straightforward calculation yields the following expression for the population gradient, which is defined as the expectation of $\nabla F(\mathbf{x})$ and corresponds to the limiting case $m = \infty$.

$$\nabla f(\mathbf{x}) = \mathbb{E}[\nabla F(\mathbf{x})] = \left( 3\|\mathbf{x}\|_2^2 - \|\mathbf{x}^\star\|_2^2 \right) \mathbf{x} - 2(\mathbf{x}^T \mathbf{x}^\star) \mathbf{x}^\star.$$

We see that, if $m = \infty$, mirror descent has three types of fixed points: a local maximum at $\mathbf{x}^{(1)} = \mathbf{0}$, saddle points at any $\mathbf{x}^{(2)}$ satisfying $\|\mathbf{x}^{(2)}\|_2^2 = \frac{1}{3} \|\mathbf{x}^\star\|_2^2$ and $(\mathbf{x}^{(2)})^T \mathbf{x}^\star = 0$, and the global minima $\mathbf{x}^{(3)} = \pm \mathbf{x}^\star$. Although the landscape might be less well-behaved if $m$ is finite, this consideration provides an intuitive explanation for the choice of initialization (5), namely, that this initialization is suitably far away from the saddle points of the population risk $f$.

Compared to initialization schemes typically employed in existing non-convex optimization based approaches to sparse phase retrieval, such as the spectral initialization used in CRAF [56] and the orthogonality promoting initialization used in SPARTA [51], the initialization we use is much simpler: this method requires only a single coordinate on the support, while the aforementioned initialization schemes require estimating the full support followed by a spectral or orthogonality-promoting scheme.

Finally, the fact that the function $f$ has saddle points means that variational coherence as defined in [59, 60] is not satisfied in our problem. Further, the results in [59, 60] are formulated for the online setting, where by design one has continuous access to independent observations, while we consider batch mirror descent with a fixed number of observations.

## 4 Main result

In this section, we present the main result of this paper. We show that continuous-time mirror descent recovers any $k$-sparse signal $\mathbf{x}^\star \in \mathbb{R}^n$ with $x^\star_{min} \geq \Omega(1/\sqrt{k})$ with high probability from $\widetilde{\mathcal{O}}(k^2)$ Gaussian measurements, where we denote $x^\star_{min} := \min_{i:x^\star_i \neq 0} |x^\star_i|/\|\mathbf{x}^\star\|_2$. Since $\mathbf{x}^\star$ cannot be distinguished from $-\mathbf{x}^\star$ using phaseless measurements, we consider, for any vector $\mathbf{x} \in \mathbb{R}^n$, the distance $\mathrm{dist}(\mathbf{x}, \mathbf{x}^\star) = \min\{\|\mathbf{x} - \mathbf{x}^\star\|_2, \|\mathbf{x} + \mathbf{x}^\star\|_2\}$. To simplify the presentation in what follows, we assume that the initialization in (5) satisfies $x_{i_0} > 0$, with which we will show convergence to $\mathbf{x}^\star$; otherwise, we can show that the algorithm converges to $-\mathbf{x}^\star$.

The following lemma characterizes the relationship between the Bregman divergence $D_\Phi(\mathbf{x}^\star, \mathbf{x})$ associated to the hypentropy mirror map (4) and the $\ell_2$ norm $\|\mathbf{x} - \mathbf{x}^\star\|_2$. For a vector $\mathbf{x} \in \mathbb{R}^n$ and a subset of coordinates $\mathcal{S} \subset \{1, ..., n\}$, we write $\mathbf{x}_\mathcal{S} = (x_i)_{i \in \mathcal{S}} \in \mathbb{R}^{|\mathcal{S}|}$.

**Lemma 1** *Let $\mathbf{x}^\star \in \mathbb{R}^n$ be any $k$-sparse vector with $x^\star_{min} \geq c/\sqrt{k}$ for some constant $c > 0$. Let $\mathcal{S} = \{1 \leq i \leq n : x^\star_i \neq 0\}$ be its support, and let $\Phi$ be as in (4) with parameter $\beta > 0$.*

- *For any vector $\mathbf{x} \in \mathbb{R}^n$, we have*

$$\|\mathbf{x} - \mathbf{x}^\star\|_2^2 \leq 2\sqrt{\max\{\|\mathbf{x}\|_\infty^2, \|\mathbf{x}^\star\|_\infty^2\} + \beta^2} \cdot D_\Phi(\mathbf{x}^\star, \mathbf{x}). \tag{7}$$

- *Let $\mathbf{x} \in \mathbb{R}^n$ be any vector with $x_i x^\star_i \geq 0$ (no mismatched sign) and $|x_i| \geq \frac{1}{2}|x^\star_i|$ for all $i = 1, ..., n$. We have*

$$D_\Phi(\mathbf{x}^\star, \mathbf{x}) \leq \frac{\sqrt{k}}{c\|\mathbf{x}^\star\|_2}\|\mathbf{x}_\mathcal{S} - \mathbf{x}^\star_\mathcal{S}\|_2^2 + \|\mathbf{x}_{\mathcal{S}^c}\|_1. \tag{8}$$

The bound in (7) shows that when a vector $\mathbf{x}$ is close to $\mathbf{x}^\star$ in terms of the Bregman divergence $D_\Phi$, then $\mathbf{x}$ is also close to $\mathbf{x}^\star$ in the $\ell_2$ sense. This means that if we are interested in convergence with respect to the $\ell_2$ norm, we can consider the Bregman divergence $D_\Phi$ as a proxy, and we write $\mathrm{dist}_\Phi(\mathbf{x}^\star, \mathbf{x}) = \min\{D_\Phi(\mathbf{x}^\star, \mathbf{x}), D_\Phi(-\mathbf{x}^\star, \mathbf{x})\}$. The bound in (8) shows that, for certain vectors $\mathbf{x} \in \mathbb{R}^n$ of interest, the Bregman divergence $D_\Phi$ can also be upper bounded in terms of a combination of the $\ell_1$ and $\ell_2$ norms. Note that, because of the assumption $|x_i| \geq \frac{1}{2}|x^\star_i|$, the bound in (8) does not depend on the parameter $\beta$. Details of the proof are provided in the full version of this paper [53].

We can now formulate our main result. The constants $c, c_1, c_2, c_3, c_4$ and $c_5$ mentioned in the following theorem are universal constants, and explicit expressions for these constants are given in the proof, which can be found in the full version of this paper [53].

**Theorem 2** *Let $\mathbf{x}^\star \in \mathbb{R}^n$ be any $k$-sparse vector with $x^*_{min} \geq c/\sqrt{k}$ for some constant $c > 0$, and let $\mathcal{S} = \{1 \leq i \leq n : x^*_i \neq 0\}$ be its support. There exist constants $c_1, c_2, c_3, c_4, c_5 > 0$ such that the following holds. Let $m \geq c_1 \max\{k^2 \log^2 n, \log^5 n\}$, and let $\mathbf{X}(t)$ be given by the continuous time mirror descent equation (1) with mirror map (4) and initialization (5) with $\beta \leq c_2\|\mathbf{x}^\star\|_2/n^4$. Let $\delta = c_3 n\sqrt{\frac{\beta}{\|\mathbf{x}^\star\|_2}}$ and $T_2(\delta) = \inf\{t > 0 : \mathrm{dist}_\Phi(\mathbf{x}^\star, \mathbf{X}(t)) \leq 2\delta\|\mathbf{x}^\star\|_2)\}$.*

*Then, with probability at least $1 - c_4 n^{-10}$, there is a $T_1(\beta) \leq c_5 k \log \frac{\|\mathbf{x}^\star\|_2}{\beta} \cdot \log(k \log \frac{\|\mathbf{x}^\star\|_2}{\beta}) \cdot \|\mathbf{x}^\star\|_2^{-3}$ such that*

$$\frac{\mathrm{dist}_\Phi(\mathbf{x}^\star, \mathbf{X}(t))}{\|\mathbf{x}^\star\|_2} \leq \frac{6\sqrt{k}}{c} \cdot \exp\left(-\frac{c\|\mathbf{x}^\star\|_2^3}{4\sqrt{k}}(t - T_1(\beta))\right) \quad \text{for all } T_1(\beta) \leq t \leq T_2(\delta). \tag{9}$$

*Further, for all $t \leq T_2(\delta)$, we have*

$$\|\mathbf{X}_{\mathcal{S}^c}(t)\|_1 \leq \delta\|\mathbf{x}^\star\|_2. \tag{10}$$

The high-level idea of the proof of Theorem 2 is as follows. First, considering the limiting case $m = \infty$, i.e. assuming we had access to the population gradient $\nabla f$, we show that mirror descent is variationally coherent along its trajectory, namely $\frac{d}{dt} D_\Phi(\mathbf{x}^\star, \mathbf{X}(t)) = \langle \nabla f(\mathbf{X}(t)), \mathbf{x}^\star - \mathbf{X}(t) \rangle < 0$. In order to prove convergence for a finite $m$, we show that, if $m \geq \widetilde{\mathcal{O}}(k^2)$, then the empirical gradient $\nabla F$ is sufficiently close to its expectation $\nabla f$ using concentration results for Lipschitz functions and bounded random variables. A detailed proof is given in the full version of this paper [53].

**Convergence of mirror descent**    The bound (9) in Theorem 2 indicates that the convergence of mirror descent (measured by the Bregman divergence $D_\Phi$) can be described as follows: in an initial warm-up period, the Bregman divergence decreases to $\sqrt{k}\|\mathbf{x}^\star\|_2$ (up to constants). Then, convergence is linear up to a precision determined by the mirror map parameter $\beta$ and the dimension of the signal $n$. This behavior can be explained as follows. The initial warm-up period is caused by the fact that, as manifested in the proof of Theorem 2, the initial Bregman divergence $D_\Phi(\mathbf{x}^\star, \mathbf{X}(0))$ scales like $\sqrt{k} \log \frac{\|\mathbf{x}^\star\|_2}{\beta}$. The following linear convergence stage corresponds to variables $X_i(t)$ on the support being fitted; to establish linear convergence, we crucially use the bound (8) of Lemma 1 along with the fact that the second term $\|\mathbf{X}_{\mathcal{S}^c}(t)\|_1$ is negligibly small compared to $\|\mathbf{X}_\mathcal{S}(t) - \mathbf{x}_\mathcal{S}^\star\|_2^2$.

**Role of the mirror map parameter**    In Theorem 2, the role of the parameter $\beta$ is to ensure that off-support variables stay sufficiently small, cf. (10). In Theorem 2, we require $\beta \leq \mathcal{O}(\|\mathbf{x}^\star\|_2/n^4)$. Note that both this requirement and the bound (10) are pessimistic and not sharp in general. The important property is that the bound on $\|\mathbf{X}_{\mathcal{S}^c}(t)\|_1$ depends polynomially on the parameter $\beta$. The price we pay for choosing a small $\beta$ is a longer warm-up period, whose length scales logarithmically in the parameter $\beta$ (see the definition of $T_1(\beta)$). In practice, we would simply choose a very small $\beta$ (e.g. $10^{-10}$), as the improvement in precision up to which we have linear convergence scales polynomially in $\beta$, while the price we pay in terms of a longer warm-up period only scales logarithmically in $\beta$. A similar trade-off between statistical accuracy and computaional cost with respect to the choice of initialization has been previously observed in [47].

**Scaling with signal magnitude**    When analyzing the convergence speed of continuous-time mirror descent equipped with the hypentropy mirror map for sparse phase retrieval, $t\|\mathbf{x}^\star\|_2^3$ is a natural quantity to consider. Recall that in sparse phase retrieval, the goal is to recover a signal $\mathbf{x}^\star$ from a set of phaseless measurements $\{(\mathbf{A}_j^T\mathbf{x}^\star)^2\}$. This problem is equivalent to the alternative problem of recovering the vector $a\mathbf{x}^\star$ from observations $\{a^2(\mathbf{A}_j^T\mathbf{x}^\star)^2\}$, for any $a \neq 0$. However, in the alternative problem, $\mathbf{X}(t)$ is not replaced by $a\mathbf{X}(t)$, and $\frac{d}{dt}\mathbf{X}(t)$ not by $a\frac{d}{dt}\mathbf{X}(t)$. If we replace $\mathbf{x}^\star$ by $a\mathbf{x}^\star$, the natural choice for the mirror map parameter becomes $a\beta$, as our results depend on the parameter $\beta$ only via the ratio $\beta/\|\mathbf{x}^\star\|_2$. Recalling the initialization (5), we see that also $\mathbf{X}(0)$ is replaced by $a\mathbf{X}(0)$ in the alternative problem. This means that, in the definition of $\frac{d}{dt}\mathbf{X}(t)$ (1), the inverse Hessian $(\nabla^2\Phi(\mathbf{X}(t)))^{-1}$ is multiplied by $a$, while the gradient $\nabla F(\mathbf{X}(t))$ is multiplied by $a^3$, cf. (6). Hence, in the alternative problem formulation, $\frac{d}{dt}\mathbf{X}(t)$ is replaced by $a^4\frac{d}{dt}\mathbf{X}(t)$, which makes $t\|\mathbf{x}^\star\|_2^3$ the right quantity to consider for the convergence speed to stay unchanged.

When considering the algorithm in discrete time, this suggests that the step size should scale like $\|\mathbf{x}^\star\|_2^{-3}$. A similar observations has been made in the case of gradient descent for phase retrieval [33], where the step size scales as $\|\mathbf{x}^\star\|_2^{-2}$. We have an extra factor $\|\mathbf{x}^\star\|_2^{-1}$ because of the mirror map.

**Remark 1 (On sample complexity)**    *Up to logarithmic term, the sample complexity in Theorem 2 matches that of existing results [10, 38, 39, 51, 55]. We typically have $k^2 \log^2 n > \log^5 n$ in regimes of interest, so that our sample complexity bound reads $\mathcal{O}(k^2 \log^2 n)$; the factor $\log^2 n$ is likely an artifact of the our proof technique, and we expect that it is possible to improve the bound to $\mathcal{O}(k^2 \log n)$. The empirical results of [54] suggest that the sample complexity of HWF, which is closely related to mirror descent as we will see in the next section, depends on the maximum signal component $\max_i |x_i^\star|/\|\mathbf{x}^\star\|_2$. The main bottleneck to establishing such a dependence in our theory seems to be the dependency between the estimates $\mathbf{X}(t)$ and the measurement vectors $\{\mathbf{A}_j\}$, and is likely to require tools different from the ones we use to prove Theorem 2.*

# 5  Connection with Hadamard Wirtinger flow

In discrete time, it has been shown that the exponentiated gradient algorithm with positive and negative weights (EG±) [28] without normalization is equivalent to mirror descent equipped with the hypentropy mirror map [22]. This equivalence has been used in [48] to recover results on implicit regularization in linear models using tools from the mirror descent literature. Since HWF performs Euclidean gradient descent on the empirical risk with Hadamard parametrization, HWF can be recovered as a discrete-time first-order approximation to the mirror descent algorithm we analyzed in Section 4. Hence, the convergence behavior established in Theorem 2 for continuous-time mirror descent might guide the development of analogous guarantees for HWF. In particular, our analysis suggests a principled approach to address the slow convergence of HWF pointed out in [54].

First, consider the following version of the exponentiated gradient algorithm in continuous time:

$$\mathbf{X}(t) = \mathbf{U}(t) - \mathbf{V}(t)$$

$$\frac{d}{dt}\mathbf{U}(t) = -\mathbf{U}(t) \odot \nabla F(\mathbf{X}(t)), \qquad \frac{d}{dt}\mathbf{V}(t) = \mathbf{V}(t) \odot \nabla F(\mathbf{X}(t)) \tag{11}$$

with initialization, writing $\hat{\theta} = (\sum_{j=1}^{m} Y_j/m)^{\frac{1}{2}}$ for the estimate of the signal size $\|\mathbf{x}^\star\|_2$,

$$U_i(0) = \begin{cases} \frac{\hat{\theta}}{2\sqrt{3}} + \sqrt{\frac{\hat{\theta}^2}{12} + \frac{\beta^2}{4}} & i = i_0 \\ \frac{\beta}{2} & i \neq i_0 \end{cases}, \qquad V_i(0) = \begin{cases} -\frac{\hat{\theta}}{2\sqrt{3}} + \sqrt{\frac{\hat{\theta}^2}{12} + \frac{\beta^2}{4}} & i = i_0 \\ \frac{\beta}{2} & i \neq i_0 \end{cases}, \tag{12}$$

where $i_0$ is defined as in (5) and the notation $\odot$ denotes the elementwise Hadamard product. Similar to the discrete case [22], a brief computation shows that the exponentiated gradient algorithm EG± (11) with initialization (12) is equivalent to mirror descent (1) with initialization (5), see the full version of this paper [53]. In particular, this reveals that the parameter $\beta$ in the hypentropy mirror map can be interpreted as the initialization size (with a factor $\frac{1}{2}$) in the exponentiated gradient formulation.

In discrete time, the exponentiated gradient algorithm (11) reads

$$\mathbf{X}(t) = \mathbf{U}(t) - \mathbf{V}(t)$$

$$\mathbf{U}(t+1) = \mathbf{U}(t) \odot \exp\left(-\eta \nabla F(\mathbf{X}(t))\right), \qquad \mathbf{V}(t+1) = \mathbf{V}(t) \odot \exp\left(\eta \nabla F(\mathbf{X}(t))\right), \tag{13}$$

with the same initialization (12), where $\eta > 0$ is the step size. Noting that $e^x \approx 1 + x$, the update (13) can be approximated by (with the step size $\eta$ rescaled by a factor 4)

$$\mathbf{X}(t) = \mathbf{U}(t) \odot \mathbf{U}(t) - \mathbf{V}(t) \odot \mathbf{V}(t)$$

$$\mathbf{U}(t+1) = \mathbf{U}(t) \odot \left(\mathbf{1}_n - 2\eta \nabla F(\mathbf{X}(t))\right), \qquad \mathbf{V}(t+1) = \mathbf{V}(t) \odot \left(\mathbf{1}_n + 2\eta \nabla F(\mathbf{X}(t))\right), \tag{14}$$

where $\mathbf{1}_n \in \mathbb{R}^n$ denotes the vector of all ones. This is exactly the update of HWF with slightly different initial values $U_{i_0}(0)$ and $V_{i_0}(0)$ in (12) compared to [54]. Note that $U_i$ and $V_i$ in (14) correspond to the square root of $U_i$ and $V_i$ in (13), respectively. The trajectory of HWF with these two initializations is essentially identical, and we only report the results using the initialization (12).

The convergence of HWF observed in [54] matches the behavior suggested by Theorem 2: first, the estimate $\mathbf{X}(t)$ barely changes (in the $\ell_2$ sense) during the initial warm-up period, which is followed by a stage of linear convergence towards the signal $\mathbf{x}^\star$, after which the convergence slows down. Further, Theorem 2 implies that the precision up to which convergence is linear is controlled by the mirror map parameter $\beta$ or, equivalently, by the initialization size in HWF. This means that, by choosing $\beta$ sufficiently small, we can avoid the final stage where convergence is slow, which is the stage mainly responsible for the high number of iterations needed to reach a given precision $\epsilon$.

In the following, we present simulations showing how the parameter $\beta$ affects the convergence of HWF. We note that the trajectories of EG± (13) and HWF (14) are essentially identical, so we only show the results for HWF. We run HWF as proposed in [54], with initialization given by the square root of the values in (12). The index $i_0$ in (12) is estimated by choosing the largest instance in $\{\sum_{j=1}^{m} Y_j A_{ji}^2\}$ as proposed in [54]. For the step size $\eta$, we follow [54] and choose $\eta = 0.1$. As discussed in Section 4, we would set $\eta = 0.1/\|\mathbf{x}^\star\|_2^3$ if $\|\mathbf{x}^\star\|_2 \neq 1$ and, if $\|\mathbf{x}^\star\|_2$ is unknown, it can be reliably estimated by $\sqrt{\frac{1}{m}\sum_{j=1}^{m} Y_j}$ [11, 50].

The setup for the simulations is as follows. We generate a 10-sparse signal vector $\mathbf{x}^\star \in \mathbb{R}^{50000}$ by first drawing $\mathbf{x}^\star \sim \mathcal{N}(\mathbf{0}, \mathbf{I}_{50000})$, then setting 49990 random entries of $\mathbf{x}^\star$ to zero, and finally normalizing

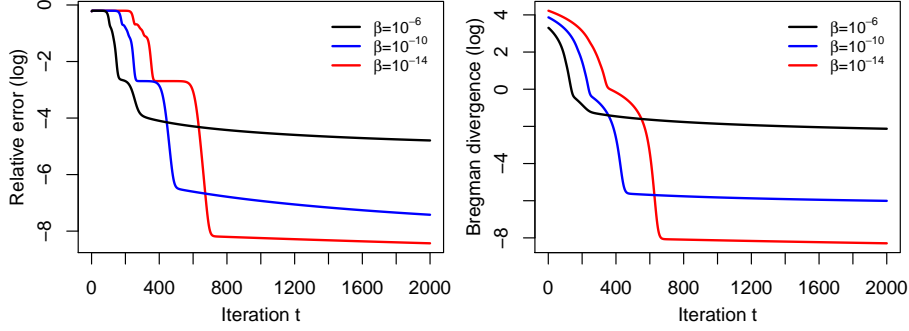

Figure 1: Convergence behavior of HWF with $n = 50000$, $m = 1000$ and $k = 10$. Left: relative error (log-scale) of HWF for $\beta = 10^{-6}$ (black line), $\beta = 10^{-10}$ (blue line) and $\beta = 10^{-14}$ (red line). Right: Bregman divergence (log-scale) for the same values of $\beta$. In all experiments, the same signal vector $\mathbf{x}^\star$ and measurement vectors $\{\mathbf{A}_j\}$ were used.

the vector to $\|\mathbf{x}^\star\|_2 = 1$. We sample $m = 1000$ Gaussian measurement vectors $\mathbf{A}_j \sim \mathcal{N}(\mathbf{0}, \mathbf{I}_{50000})$ and generate phaseless measurements $Y_j = (\mathbf{A}_j^T \mathbf{x}^\star)^2$.

We evaluate the relative error as well as the relative Bregman divergence from the solution set $\{\pm \mathbf{x}^\star\}$, given by

$$\frac{\text{dist}(\mathbf{X}(t), \mathbf{x}^\star)}{\|\mathbf{x}^\star\|_2}, \qquad \text{and} \qquad \frac{\text{dist}_\Phi(\mathbf{x}^\star, \mathbf{X}(t))}{\|\mathbf{x}^\star\|_2}.$$

Figure 1 (left) shows that HWF exhibits the behavior suggested by our theory, namely that, after an initial warm-up stage, convergence is linear up to a precision depending on $\beta$. As we decrease the parameter $\beta$ from $10^{-6}$ to $10^{-14}$, the length of the initial warm-up stage increases as $\log \frac{1}{\beta}$. Note that in this first stage we see repeated drops and plateaus. The use of the hypentropy mirror map means that coordinates $X_i(t)$ that are small in magnitude can only change slowly, as then also the inverse Hessian $(\nabla^2 \Phi(\mathbf{X}))_{ii}^{-1} = (X_i^2 + \beta^2)^{\frac{1}{2}}$ is small, cf. (1). Informally, each drop corresponds to one coordinate $X_i(t)$ becoming large, while each plateau corresponds to all "large" coordinates reaching a stable state, at which point $\mathbf{X}(t)$ barely moves as the other "small" coordinates only change very slowly. This is followed by a second stage where the error decreases linearly up to a precision which depends polynomially on $\beta$ (note the log-scale of the $y$-axis). Finally, the convergence slows down after this precision has been reached. In order to reach a given precision $\epsilon > 0$, it is preferable to choose the parameter $\beta$ sufficiently small so that convergence is linear up to precision $\epsilon$. This way, we avoid the final slow convergence stage, while the number of iterations spent in the initial warm-up stage only increases as $\log \frac{1}{\beta}$.

The right plot of Figure 1 shows a similar behavior when we consider the Bregman divergence $D_\Phi(\mathbf{x}^\star, \mathbf{X}(t))$. The only difference is in the first stage, where the Bregman divergence decreases at a constant rate without plateaus. As $\beta$ decreases, more iterations are spent in the first stage because the initial Bregman divergence $D_\Phi(\mathbf{x}^\star, \mathbf{X}(0))$ increases like $\log \frac{1}{\beta}$.

## 6   Conclusion

We provided a convergence analysis of continuous-time mirror descent applied to sparse phase retrieval. We proved that, equipped with the hypentropy mirror map, mirror descent recovers any $k$-sparse signal $\mathbf{x}^\star \in \mathbb{R}^n$ with $x_{min}^\star = \Omega(1/\sqrt{k})$ from $\widetilde{\mathcal{O}}(k^2)$ Gaussian measurements. This yields a simple algorithm, which, unlike most existing methods, does not require thresholding steps or added regularization terms to enforce sparsity. Further, as HWF can be recovered as a discrete-time first-order approximation to the mirror descent algorithm we analyzed, our results provide a principled theoretical understanding of HWF. In particular, our continuous-time analysis suggests how the initialization size in HWF affects convergence, and that choosing the initialization size sufficiently small can result in far fewer iterations being necessary to reach any given precision $\epsilon > 0$. We leave a full theoretical investigation of HWF, with a proper discussion on step-size tuning, for future work.

## Acknowledgments

Fan Wu is supported by the EPSRC and MRC through the OxWaSP CDT programme (EP/L016710/1).

## Broader impact

This work does not present any foreseeable ethical or societal consequences.

## Footnotes

[1]For a full version of this paper see [53].

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
