[Reviews · NeurIPS 2020]

Review 1

Summary and Contributions: This paper proposes a new algorithm for sparse phase retrieval, which involves recovering a k-sparse signal from squared linear measurements. Specifically, the algorithm comprises (continuous-time) mirror descent with the hypentropy mirror map, initialized away from zero. This algorithm requires no tuning parameters (beyond a temperature parameter that can be estimated fairly robustly from the measurements). The authors prove that this algorithm provably recovers any signal with approximately uniform-magnitude coefficients --- despite the fact that no sparsity regularization or projection is needed. In this sense, the mirror descent algorithm automatically adapts to the sparsity level. The authors also provide several synthetic simulation results that support the analysis. In particular, the convergence of the algorithm (measured in terms of Bregman divergence) is theoretically broken down into three phases; simulations show that this is not merely an artifact of the the proof techniques, and that mirror descent indeed exhibits such convergence behavior.

Strengths: + This paper is excellently written. + The theoretical contributions are clear and compelling. To my knowledge, this is the first algorithm on sparse phase retrieval that succeeds without any parameter tuning, which can be a significant hurdle. + The proofs involve several interesting new elements that I have not seen before in the phase retrieval literature. Among these include: use of non-Euclidean gradient flow techniques (i.e., continuous time mirror descent); use of the Bregman divergence of the mirror map to characterize convergence; careful analysis of on-support and off-support convergence; careful choice of the temperature parameter in mirror descent; connections to exponentiated gradient descent (EG-plus-minus); and many others. + This paper also fills in a few gaps in the analysis offered by the recent preprint [53] that also attempts to solve sparse phase retrieval using (vanilla) gradient descent with an overparameterized representation. + Big picture-wise, the paper adds to the growing body of literature on implicit regularization achieved by gradient-based dynamics. Sparse phase retrieval is a "prototypical" non-convex problem, in that the objective function has exponentially many local minima, and that the constraints are non-differentiable. Understanding the behavior on gradient flow-type techniques on such problems can be very useful for more general machine learning problems relevant to the NeurIPS community.

Weaknesses: [The following weaknesses can be major or minor, depending on how one interprets their relative significance. I personally think all these points are minor and should not detract from the contributions of the paper.] - The absence of noise in the measurement model is somewhat limiting. What would happen to the performance of mirror descent in such cases? I suspect the method might become somewhat unstable since mirror descent requires computing the inverse of the metric tensor in each iteration. - Practical aspects (such as how to properly discretize the continuous time updates) are not addressed. - The analysis succeeds only for signals with (approximately) similiar-magnitude coefficients.

Correctness: I believe the claims are correct, although I did not check completely. The experimental results are limited to small synthetic examples but evocative of the theoretical analysis.

Clarity: The paper is very well-written.

Relation to Prior Work: The paper does a very good job of connecting to existing work in both the phase retrieval as well as the algorithmic machine learning literature.

Reproducibility: Yes

Additional Feedback:


Review 2

Summary and Contributions: The authors consider the sparse (real) gaussian phase retrieval (GPR) problem, with mean squared error objective and m square measurements of a k-sparse n-dimensional signal. Their main contribution is a full convergence analysis for a continuous-time mirror descent algorithm for this objective, using the hypentropy mirror map and an initialization scheme that requires knowledge of one coordinate of the ground truth's support (Theorem 2): they prove that when the ell^2-normalized minimum nonzero element of the ground truth vector has magnitude \Omega(1/sqrt(k)), the number of samples satisfies m >= \max{k^2, \log^3 n} \log^2 n, and the mirror descent trajectory X(t) is bounded, the trajectory X(t) converges in ell^2 to the ground truth at a rate that is initially linear, and eventually sublinear, with constants depending on k and n; the ultimate accuracy and length of the linear convergence phase can be controlled through the mirror descent parameter \beta, which is the algorithm's only hyperparameter. In reasonable scaling regimes for the sparsity k, this result matches the sample complexities of other algorithms for sparse GPR. The authors' analysis is motivated by a connection to the recently-proposed Hadamard Wirtinger flow (HWF) algorithm, a practical algorithm for sparse GPR that potentially offers increased adaptivity relative to other algorithms for sparse GPR, as it does not enforce sparsity of the signal through explicit regularization or hard-thresholding steps, and has been shown empirically to offer performance advantages in terms of signal recovery rate over other sparse GPR algorithms [53]. In particular, the authors give calculations (section 5) that show HWF can be interpreted as a first-order approximation to an algorithm equivalent to continuous-time mirror descent on the sparse GPR objective with hypentropy mirror map, and they illustrate the fruitfulness of this connection by presenting an experiment that demonstrates intuitions obtained from the mirror descent convergence theorem in the context of the HWF algorithm (after initializing HWF in a way inspired by the theorem). Besides this motivation, the authors' analysis contributes to the growing literature on gradient flow analyses for nonconvex problems, and in particular mirror descent analyses for nonconvex problems.

Strengths: - The authors analyze a conceptual algorithm for sparse GPR that has desirable qualitative features, as in HWF [53]: namely adaptivity and fewer turning parameters, with those present having simple interpretations. The convergence analysis is complete, with almost all features one would desire for an analysis of this type: the sample complexity matches that of standard sparse GPR algorithms, the hypotheses are not overly restrictive, and the aspects of the convergence behavior established that are not straightforward (i.e., the switching of the rate from linear to sublinear) are explained at a quantitative level and a qualitative level through the experiments. - The authors provide a concrete connection to HWF (section 5), which could enable the techniques underlying the convergence analysis of the present work to be applied to a similar convergence analysis of HWF. The experiments here make the point well, since the role of \beta in the convergence analysis of Theorem 2 appears to express itself identically when HWF is initialized with a mirror-descent-motivated setting of the initialization weight (this kind of understanding appears to be absent from the setting of the \alpha parameter in [53], for example). - The analysis and result of Theorem 2 lead to a useful qualitative ``three stage'' interpretation of the convergence process that seems suggestive of some of the convergence behavior of HWF observed in [53]: as described in lines 216-226, the convergence schedule goes from a slow ``warm-up'' phase to a linear convergence phase to a sublinear phase, and the authors use the analysis underlying the convergence result to give concrete reasons for the behavior in each of these stages. - Although the sublinear rate of convergence `in the tail' suggested by Theorem 2 seems pessimistic relative to the linear rates of convergence guaranteed for other sparse GPR algorithms (e.g. SPARTA), the authors also discuss (lines 227-236) the interplay between the convergence process and the setting of the mirror descent parameter \beta, demonstrating that with proper setting of \beta, one can essentially guarantee via Theorem 2 linear convergence to a desired accuracy (in particular, by making \beta very small, as the authors discuss). As in the previous item, these considerations are displayed in a small experiment in section 5 (Figure 1), which echoes the ideas the authors discuss. - The paper contributes to the general understanding of the performance of continuous time mirror descent on nonconvex optimization problems with statistical structure; understanding the technical conditions necessary for success in such settings, as advanced here, may be of general use.

Weaknesses: - The continuous-time mirror descent algorithm studied here is not a practical algorithm for sparse GPR, since in practice one would have to work with a discretization, and the analysis only applies to the continuous time setting. However, this type of analysis is becoming more standard recently, and can be valuable for getting algorithmic insight in a setting with fewer technical difficulties (as the authors discuss in the introduction). This notion is borne out convincingly in the calculations experiments of section 5, which transfer some insights from the convergence analysis of Theorem 2 to the empirical performance of HWF. - It is not a weakness per se, but a point where connections to the intuitions behind improved recovery performance of HWF in certain sparsity regimes (namely, regimes where there are a small number of entries in the ground truth x* that are as large as a constant multiple of the ell^2 norm) demonstrated empirically in [53] are missed: as the authors discuss in Remark 1, the sample complexity in Theorem 2 goes to O(k^2) instead of involving k || x* ||_2^2 || x* ||_{\infty}^{-2} (ignoring logarithmic factors) as the HWF experiments suggest should be the case, which precludes an improved sample complexity in the aforementioned setting.

Correctness: - Due to the length of the appendix I have not been able to review all of the proofs of the supporting lemmas for Theorem 2 line by line. Based on the authors' summary of the main ideas of the proof and my understanding of prior works with a somewhat similar flavor (e.g. Sun et al. (2017), Chen et al. (2019)), I believe the high-level approach and tools should be commensurate with the task of establishing Theorem 2.

Clarity: - The paper is well written and organized; the authors' writing makes it is easy to understand the mirror descent algorithm, the initialization scheme, and other relevant technical details. Examples and intuitions (e.g. sections 4-5) are pertinent and aid in appreciation of subtleties of the material. - Small potential typo issues: - - (7) and (8) involve the Bregman divergence of \psi instead of \Phi? - - I may be missing something (per the discussion in ``Scaling with signal magnitude'' on page 6), but is the definition of x^{\star}_{min} in line 181 supposed to be normalized by || x^{\star} ||_2? This seems to clash with the normalization that is used in the lower bounds on x^{\star}_{min} in the first lines of the statements of Lemma 1 and Theorem 2; but I may be missing something here.

Relation to Prior Work: - The paper is well-referenced, with an appropriate discussion of contemporary works on algorithms for sparse phase retrieval (and how they differ, in terms of advantages/disadvantages, from HWF/the present work's mirror descent algorithm), and technical approaches used in similar works on analyzing mirror descent for nonconvex problems, in particular exploiting the presence in this problem of an ``along-trajectory'' variational coherence property is likely similar to the kinds of analyses used to prove first-order methods succeed in similar problems (e.g. matrix sensing/factorization).

Reproducibility: Yes

Additional Feedback: --- POST-REBUTTAL --- I thank the authors for their response to my review, in particular for their insightful response to my fairly vaguely-worded thoughts on the sample complexity in the main theorem. After reading the other referees' reviews, I have decided to keep my score the same. I am eager to see what theoretical work will follow from the authors' result. ---------------------


Review 3

Summary and Contributions: This paper studies the problem of sparse phase retrieval. For that a new approach based on mirror-descent (using the hypertrophy mirror map) is used. The authors prove that with at least order of k^2 measurements their (continuous-time) algorithm recovers the unknown signal (ignoring log-factors and k denotes the sparsity of the signal.) Furthermore, the claims are supported by experiments and the connection to the "Hadamard Wirtinger Flow" is discussed. ------------------- After reading the rebuttal as well as the other referee's comments, I changed my opinion on the strength of the paper. For this reason, I will change my score to 7.

Strengths: Sparse phase retrieval is an important problem to both the machine learning and the signal processing community. This paper provides a new approach together with a theoretical analysis. I think that this new point view and the connection between mirror descent, Hadamard Wirtinger Flow, and sparse phase retrieval is inspiring and will trigger follow-up work. In addition, this paper might help in understanding non-convex problems and mirror descent for those problems in general.

Weaknesses: -For the main result, it is required that x_min is at least at the order of ||x||/sqrt(k). This is a fairly strong additional assumption and it is also very unintuitive: Signals where the mass is equidistributed over the support should be harder to recover than signals which are largely unbalanced. The paper would be stronger if this assumption would have been relaxed. Note, for example, [10] does not need this assumption. I would ask the authors, in Remark 1, when citing this paper, to also admit this and not say "the sample complexity in Theorem 2 matches that of existing results". Moreover, the authors might point the interested reader to the exact location in the proof where this assumption is needed. (While strengthening the result may not be possible during the rebuttal phase, I would ask the authors to consider the points listed in the section above.) -Regarding the experiment in Section 5: The authors have chosen m=n=1000. This means the number of measurements is already as large as the ambient dimension. In my own experience, this can lead to many artifacts. The experiments would be much more convincing, if the authors would devise a setting where k^2 <m << n.

Correctness: The claims and proofs in the paper are sound.

Clarity: The paper is well written and very clear.

Relation to Prior Work: In general, prior work is discussed with sufficient detail. However, the authors might add the following references which deal with sparse phase retrieval: -"Structured Signal Recovery From Quadratic Measurements: Breaking Sample Complexity Barriers via Nonconvex Optimization", Soltanolkotabi https://ieeexplore.ieee.org/document/8606215 (solves phase retrieval with optimal sample complexity via projected gradient descent given a good initialization) -"Solving Equations of Random Convex Functions via Anchored Regression" Bahmani, Romberg https://link.springer.com/article/10.1007/s10208-018-9401-4 (solves sparse phase retrieval with optimal sample complexity given a good initialization)

Reproducibility: Yes

Additional Feedback: -Appendix, p. 14: In the last equation of the proof of Lemma 1, a factor 2 is missing after the integration.

[Author Response · NeurIPS 2020]

We thank the reviewers for their feedback.

**On the role of $x^\star_{min}$ (R1, R3)**

We expect that the assumption $x^\star_{min} \geq \Omega(\|\mathbf{x}^\star\|_2/\sqrt{k})$ (where $x^\star_{min} = \min_{i:x^\star_i \neq 0} |x^\star_i|$) is likely an artifact of our proof.
However, we expect a proof without this assumption (if feasible) to be more complicated, possibly distracting from the
main ideas of the (already lengthy) proof. On the experimental side, both our setting and the setting considered in [53]
uses Gaussian signals without any restriction on $x^\star_{min}$, which might indicate that the assumption on $x^\star_{min}$ is in fact not
necessary for mirror descent to reconstruct the signal $\mathbf{x}^\star$.

On the theoretical side, the assumption on $x^\star_{min}$ appears in two places in our analysis. On a high level, the inner product
$\mathbf{X}(t)^T\mathbf{x}^\star$ is a key quantity in showing the convergence of mirror descent. Using the fact that we have no mismatched
signs (Lemma 5 equation (20)), we have the simple lower bound $|\mathbf{X}(t)^T\mathbf{x}^\star| = |\sum_{i=1}^n X_i(t)x^\star_i| \geq \|\mathbf{X}(t)_\mathcal{S}\|_1 x^\star_{min}$,
where $\mathcal{S} = \{i : x^\star_i \neq 0\}$ denotes the support of the signal. Our technical lemmas then guarantee that the discrepancy
between the empirical gradient $\nabla F$ and the population gradient $\nabla f$ is sufficiently small compared to $\mathbf{X}(t)^T\mathbf{x}^\star$. We
believe that it might be possible to control the inner product via a more refined analysis of the trajectory of mirror
descent instead of assuming $x^\star_{min} \geq \Omega(\|\mathbf{x}^\star\|_2/\sqrt{k})$, however it is likely to require different techniques from the ones
used in our analysis. Second, the assumption on $x^\star_{min}$ allows us to separate *all* support coordinates from off-support
coordinates at the end of the initial warm-up stage, in the sense that $|X_i(t)| \gg |X_j(t)|$ for all $i \in \mathcal{S}$, $j \notin \mathcal{S}$. Intuitively,
we neither expect nor need $|X_i(t)| > |X_j(t)|$ if $|x^\star_i|$ is very small. Rather, it should suffice if above inequality holds
for $i \in \mathcal{S}$ corresponding to "large" coordinates. We anticipate that it might be possible to make this intuition rigorous,
potentially utilising similar tools as the ones used in [10] to eliminate the need for an assumption on $x^\star_{min}$.

**Experiment in a setting with $k^2 < m \ll n$ (R3)**
Following the reviewer's suggestion, we repeated the experiment of Section 5 in various settings with $k^2 < m \ll n$.
We present an example in Figure 2, where we increased the dimension of the signal to $n = 50000$ and kept everything
else as described in Section 5. We observe the same qualitative behaviour as in Figure 1 (we only include the relative $\ell_2$
error, as the Bregman divergence also shows the same behaviour as in Figure 1).

**On noise in the measurement model (R1)**
We ran discrete-time mirror descent in a measurement model with additive white Gaussian noise, $Y_j = (\mathbf{A}_j^T\mathbf{x}^\star)^2 + \varepsilon_j$
where $\varepsilon_j \sim \mathcal{N}(0,\sigma^2)$ i.i.d. for some $\sigma^2 > 0$. We show the results of an experiment with $n = 50000$, $m = 1000$,
$k = 10$ and $\sigma^2 = 0.1$ below. Figure 3 suggests that mirror descent can also reconstruct sparse signals in the model
with noise, and the parameter $\beta$ seems to affect convergence in a similar way as in the noiseless case. The precision
up to which we have linear convergence barely improves as we decrease $\beta$ from $10^{-10}$ to $10^{-14}$, which we suspect
is because the presence of noise in the measurement model limits the attainable accuracy. In all our experiments, the
relative $\ell_2$ error increases after reaching a minimum, which suggests that additional techniques such as early stopping
might be needed. We leave this to future work, as the analysis of the noisy model is likely to involve novel ideas.

**On the improved sample complexity of HWF (R2)**
Theorem 2 requires the number of measurements $m$ to be of order $k^2$ (ignoring logarithmic factors). The empirical
results in [53] suggest that HWF is able to reconstruct sparse signals from far fewer measurements ($m < k^2$) if the
signal contains one large entry. There are two obvious candidate explanations for this discrepancy: it could be the case
that 1) our proof is suboptimal and the sample complexity in Theorem 2 is overly pessimistic, or that 2) the statement of
Theorem 2 does not hold if $m < k^2$, regardless of the maximum magnitude entry of the signal (Theorem 2 not only
guarantees convergence towards the underlying signal, but also characterizes the speed of convergence). To investigate
which of the two explanations seems more likely, we consider an experiment with $n = 1000$, $m = 500$, $k = 100$ and
one entry of the signal set to 0.7. In this setting we have $m < k^2$, and the assumptions of Theorem 2 are not satisfied.
Figure 4 suggests that explanation 2) seems more likely: while we have convergence towards the underlying signal, the
convergence behaviour is not as described by Theorem 2. In particular, we do not observe linear convergence up to a
precision depending on $\beta$.

Figure 2: Relative $\ell_2$ error (log-scale) of HWF for $n = 50000$, $m = 1000$ and $k = 10$ (no noise).

Figure 3: Relative $\ell_2$ error (log-scale) of HWF for $n = 50000$, $m = 1000$, $k = 10$ and $\sigma^2 = 0.1$.

Figure 4: Relative $\ell_2$ error (log-scale) of HWF for $n = 1000$, $m = 500$, $k = 100$ and one entry of $\mathbf{x}^\star$ set to 0.7 (no noise).

[Meta-Review · NeurIPS 2020]

All the three reviewers rated this paper favorably, well above the acceptance threshold. They agreed that the proposed method, unconstrained mirror descent in continuous time with the mirror map (4) and the initialization (5), is interesting and significant in several respects: it is almost parameter-free, the recovery guarantee has been given (Theorem 2), theoretical prediction of the three-stage behavior was confirmed empirically, and an interpretation of the Hadamard Wirtinger flow algorithm as a discrete-time first-order approximation of the proposal was provided. They also expect that this study, being inspiring, would stimulate follow-up work. I am thus glad to recommend acceptance of this paper for presentation at the NeurIPS conference.